# Prediction of Compression Index of Fine-Grained Soils Using a Gene Expression Programming Model

**Danial Mohammadzadeh S. [1,2]**, **Seyed-Farzan Kazemi [3]**, **Amir Mosavi [4,5,\*]**,
**Ehsan Nasseralshariati [6]** and **Joseph H. M. Tah [4]**

1. Department of Civil Engineering, Ferdowsi University of Mashhad, Mashhad 9177948974, Iran; d.mohammadzadeh.Sh@gmail.com
2. Department of Elite Relations with Industries, Khorasan Construction Engineering Organization, Mashhad 9185816744, Iran
3. Michael Baker International, Hamilton, NJ 08619, USA; seyedfarzan.kazemi@mbakerintl.com
4. School of the Built Environment, Oxford Brookes University, Oxford OX3 0BP, UK; jtah@brookes.ac.uk
5. Kalman Kando Faculty of Electrical Engineering, Obuda University, 1034 Budapest, Hungary
6. Department of Civil Engineering, School of Engineering, Hakim Sabzevari University, Sabzevar 980571, Iran; ehsan.shariati1990@gmail.com
* Correspondence: a.mosavi@brookes.ac.uk

**Abstract:** In construction projects, estimation of the settlement of fine-grained soils is of critical importance, and yet is a challenging task. The coefficient of consolidation for the compression index ($C_c$) is a key parameter in modeling the settlement of fine-grained soil layers. However, the estimation of this parameter is costly, time-consuming, and requires skilled technicians. To overcome these drawbacks, we aimed to predict $C_c$ through other soil parameters, i.e., the liquid limit ($LL$), plastic limit ($PL$), and initial void ratio ($e_0$). Using these parameters is more convenient and requires substantially less time and cost compared to the conventional tests to estimate $C_c$. This study presents a novel prediction model for the $C_c$ of fine-grained soils using gene expression programming (GEP). A database consisting of 108 different data points was used to develop the model. A closed-form equation solution was derived to estimate $C_c$ based on $LL$, $PL$, and $e_0$. The performance of the developed GEP-based model was evaluated through the coefficient of determination ($R^2$), the root mean squared error ($RMSE$), and the mean average error ($MAE$). The proposed model performed better in terms of $R^2$, $RMSE$, and $MAE$ compared to the other models.

**Keywords:** soil compression index; fine-grained soils; gene expression programming (GEP); prediction; big data; machine learning; construction; infrastructures; deep learning; data mining; soil engineering; civil engineering

---

## 1. Introduction

Soil compressibility is considered to be the volume reduction under load of pore water drainage. A precise estimation of this property is critical for calculating the settlement of soil layers [1]. This problem has become more critical for fine-grained soils due to their low permeability, resulting in the compression index ($C_c$) being the most accepted parameter to date to represent soil compressibility [2]. This parameter is often utilized for measuring the individual soil layer settlement. Different empirical equations have been particularly developed to predict $C_c$ [3–9]. These equations were mainly developed based on traditional statistical analyses. Nevertheless, they include a number of drawbacks, such as a low correlation between input and output parameters [10]. Thus, it is essential to develop a comprehensive model to analyze the complex behavior of $C_c$. This model should significantly eliminate

the shortcomings of the previous models, such as practicality and a low correlation between input and output parameters.

Soft computing techniques such as artificial neural networks (ANNs) are widely accepted and popular, along with conventional statistical methods (e.g., regressions) [11–21]. These techniques have been successfully applied to different geotechnical problems, such as $C_c$ prediction [7,22–27]. However, a major limitation of common soft computing techniques is that no closed-form prediction equation is provided by them. With the introduction of artificial intelligence (AI) techniques and particularly genetic programming (GP), researchers in the field of soft computing have attempted to solve this issue (i.e., obtaining a closed-form solution). AI includes various techniques of ANNs, neuro-fuzzy neural networks (ANFIS), and support vector machines (SVMs), with a great record of successful application [28,29]. With AI, a learning mechanism often contributes to constructing the intelligent structure of an estimation model. Among the popular AI methods, ANNs present a robust artificial tool that is widely used to predict $C_c$ [7,22–26]. AI techniques have been reported to have an acceptable statistical performance in terms of correlation. These techniques are often known as black box models in soft computing, and they mainly lack capability in offering closed-form estimation formulas [10]. This, been reported to be a drawback to AI techniques that limits their practicality [10,28]. Nevertheless, the runtime for most soft computing techniques could be efficiently decreased by using parallel processing methods [30]. Mohammadzadeh et al. (2014) reviewed state-of-the-art soft computing models and proposed multi-expression programming (MEP) to model the $C_c$ of fine-grained soils, and the proposed model outperformed ANNs [29].

Genetic programming (GP) and also multigene genetic programming (MGGP), which is an enhanced variation of GP using classical regression, have been used for modeling purposes (of $C_c$) [28]. Mohammadzadeh et al. (2016) built an MGGP model to estimate $C_c$ with higher accuracy, which presented promising results [28]. The GP-based methods of modeling are classified as individual computational programming, which is a major family of soft computing techniques. GP models can empower and enable complex and highly nonlinear prediction modeling tasks [31]. While classical GP nominates only a single program, gene expression programming (GEP) includes several genes of programming for reaching optimal solutions [32]. The application of GEP is growing significantly compared to GP in the engineering domain mainly due to the accuracy of its predictions [28,29]. The current study investigated the use of GEP to develop a prediction equation for the $C_c$ of fine-grained soils existing in northeastern Iran. The objective of this study was developing a GEP-based prediction equation for the $C_c$ of fine-grained soils with simple tests such as the Atterberg liquid limit (*LL*) and plastic limit (*PL*). Since conventional consolidation tests of fine-grained soils (e.g., the oedometer test) are time-consuming and costly, the application of such a prediction equation will lead to substantial savings for $C_c$ estimation in terms of cost and time.

## 2. GEP

There are several variants of GP available for modeling. GEP is the latest variant of GP, and it is a powerful tool for approximating the solution of a problem in a closed-form format. Conventional GP generates computational models through mimicking the biological evolution of living organisms, providing a tree-like form of solution, which leads to the closed-form solution of the optimization problem [28,29,31–33]. The main objective of GP is obtaining programs that connect inputs to outputs for each data point, creating a population of programs. The population of programs (in the form of a tree branch shape) created by GP includes functions and terminals, which are randomly generated. The final solution of the problem is determined based on the tree-like programs.

The foundation of modeling with GEP was first developed by Ferreira in 2002 [34] and consists of a number of components, i.e., a terminal set, a function set, control parameters, a fitness function, and a termination function. GEP employs a fixed length of character strings to model the problem, unlike the conventional GP. These characters further turn into parse trees in various sizes and shapes, known as expression trees (ETs). The benefit of GEP over conventional GP is that genetic diversity is represented

as genetic operators of chromosomes. GEP, in fact, evolves a number of genes (subprograms) [34] that are individual tree-like programs [10,34]. Furthermore, GEP has a flexible multigenetic nature suitable for the construction and evolution of complex networks of genes. In the GEP framework, the genes in a chromosome may consist of two types of information stored in either the tail or head of genes, i.e., information for generating the overall GEP model and the information from terminals for producing subsequent of the model. Specific details about GEP can be found elsewhere [10,31,32,34,35].

Figure 1 presents a sample program illustration of evolving GEP, where $d_1$, $d_2$, and $d_3$ are the model inputs. Furthermore, the process evolution functions are +, −, ×, /, exponential function (exp()), natural logarithm function (ln()), and Inv. The presented model is linear, with coefficients $c_0$, $c_1$, and $c_2$, while utilizing nonlinear terms [31,32]. For obtaining $c_0$, $c_1$, and $c_2$, a simple least square was applied to the training data. A partial least squares method could also be employed for this objective [18,22]. The important GEP parameters that need to be carefully selected are the tree depth and the quantity of genes. However, minimizing the tree depth generally results in shorter closed-form equations with fewer numbers of terms [29,34].

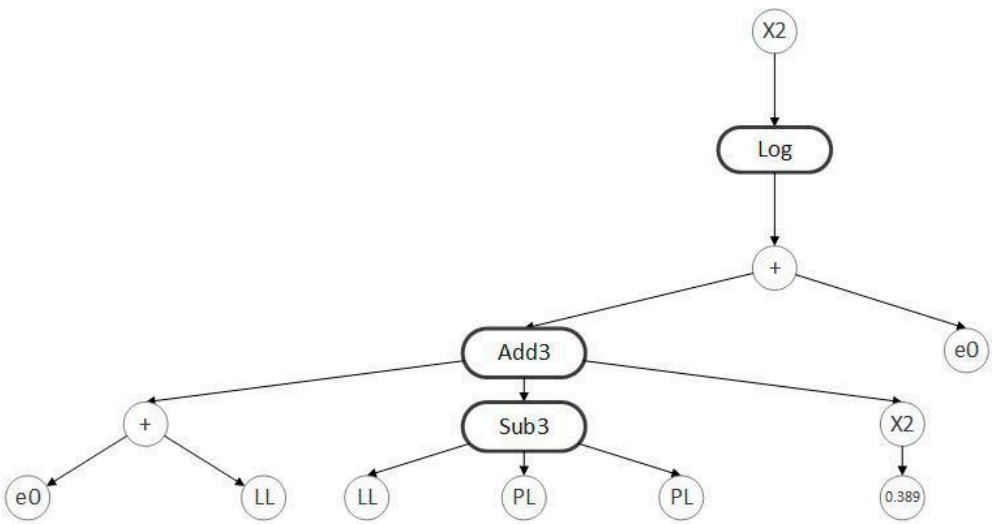

**Figure 1.** Sample gene expression programming (GEP) model.

## 3. Modeling of $C_c$ for Fine-Grained Soils

### 3.1. Data Collection

A set of 108 individual consolidation test results obtained from laboratory tests were used to develop the GEP-based prediction equation. As mentioned earlier, the objective of this study was to predict $C_c$ using conventional parameters of fine-grained soils, namely *PL*, *LL*, and $e_0$. Here, 101 out of 108 data points corresponded to test results conducted on soil samples collected from different locations in Mashhad, Iran. Soil samples were classified as silty–clayey sand (SC–SM), gravelly lean clay with sand (CL), and silty clay with sand (CL–ML) based on the unified soil classification system. These samples were cored from a depth of 0.5 m to 1.0 m. *LL*, *PL*, and $e_0$ were measured for these samples in a laboratory based on ASTM D4318-17 and ASTM D854-14 [36,37]. Furthermore, $C_c$ was measured using an oedometer test based on ASTM D2435-11 [38]. In addition, seven consolidation test results conducted by Malih [39] were integrated into the laboratory database to make it more robust. The descriptive statistics of influential input parameters (i.e., *LL*, *PL*, and $e_0$) and the output parameter, i.e., $C_c$, based on the database utilized for our study is presented in Table 1. Furthermore, Figures 2–5 illustrate the distribution of these parameters using histograms.

**Table 1.** Descriptive statistics for input and output parameters used in the GEP-based developed model. LL: liquid limit; PL: plastic limit.

| Parameter | LL (%) | PL (%) | $e_0$ | $C_c$ |
|---|---|---|---|---|
| Mean | 36.16 | 22.61 | 0.75 | 0.17 |
| Standard Deviation | 12.79 | 5.64 | 0.12 | 0.05 |
| Minimum | 19.40 | 14.80 | 0.51 | 0.08 |
| Maximum | 72.00 | 44.00 | 1.03 | 0.025 |
| Range | 52.60 | 29.20 | 0.52 | 0.18 |

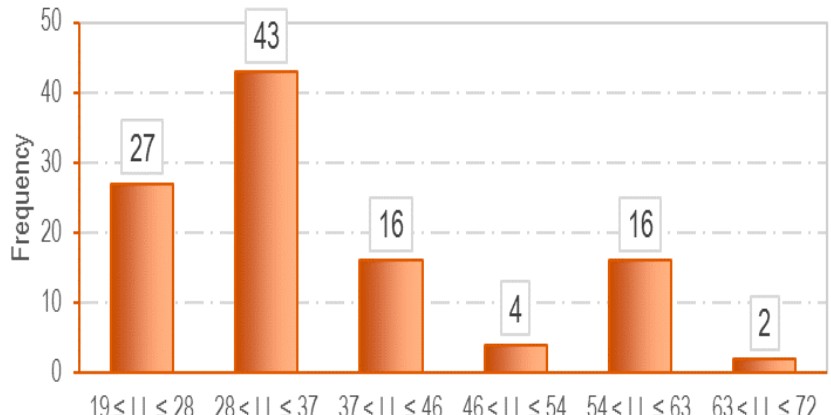

**Figure 2.** Distribution of *LL*.

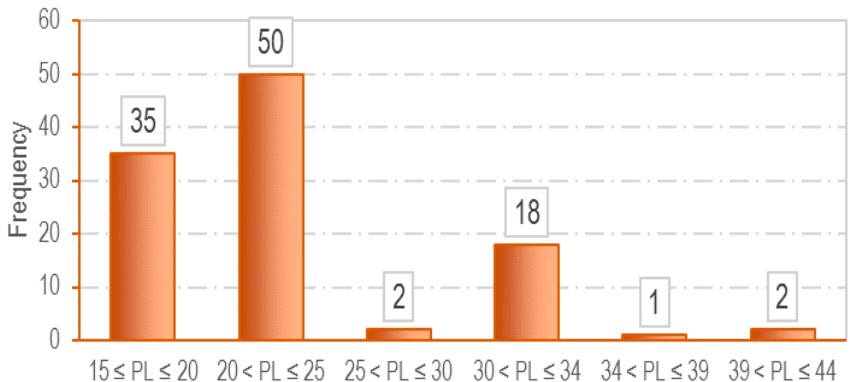

**Figure 3.** Distribution of *PL*.

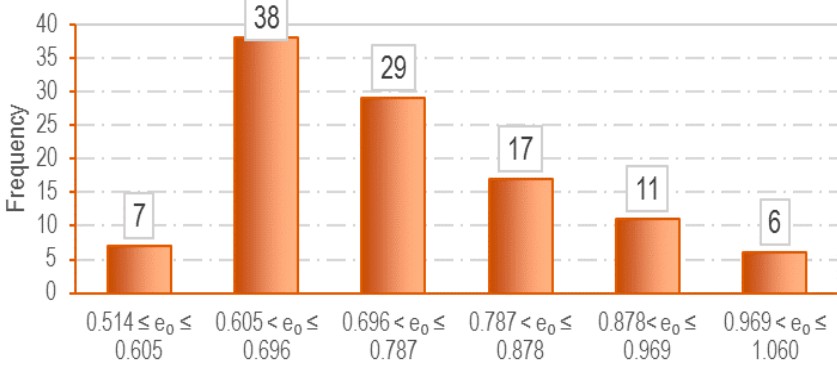

**Figure 4.** Distribution of $e_o$.

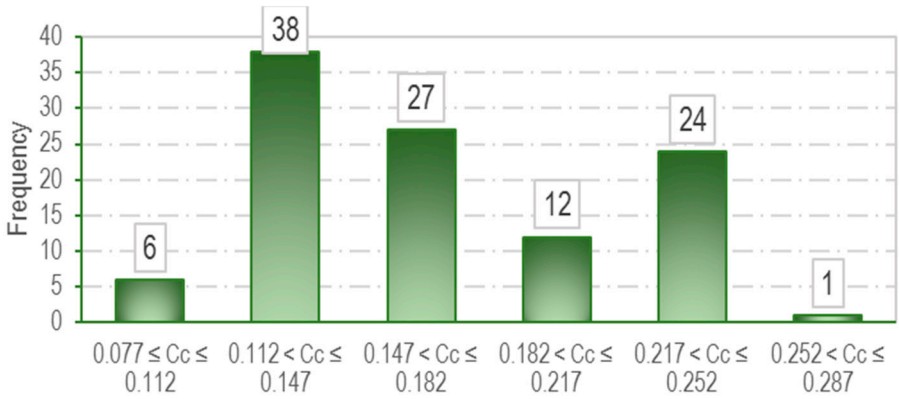

**Figure 5.** Distribution of $C_c$.

*3.2. Model Structure and Performance*

The *LL* and *PL* represent the two various states of the soil depending on its water content. The $e_0$ of soil represents the initial ratio of the volume of voids to the solids. Prediction equations for $C_c$ developed by previous studies (see Equation (1)) have clearly indicated that *LL*, *PL*, and $e_0$ are the three main parameters that influence $C_c$ [3–9]. Thus, these parameters were considered in the current study to develop a simplified prediction equation for $C_c$. The main motivation of developing such an equation was that determination of *LL*, *PL*, and $e_0$ is straightforward compared to performing any consolidation test that directly determines $C_c$. Therefore, the developed model is anticipated to result in considerable savings in terms of testing time, technician costs, and laboratory equipment. It should be noted that *LL*, *PL*, and $e_0$ are influenced by the natural water content of partially saturated soils, thus making the developed equation applicable to any saturated fine-grained soils [28,39,40]. Mathematically, the developed equation has the following structure:

$$C_c = f(LL, PL, e_0), \tag{1}$$

showing that $C_c$ is considered to be a function of *LL*, *PL*, and $e_0$. In order to develop the GEP-based prediction equation for $C_c$, a database containing 108 data points was developed. Each data point corresponded to *LL*, *PL*, and $e_0$, as well as $C_c$, for a particular fine-grained soil sample. GeneXproTools 5.0 was used to develop the GEP-based prediction equation in MATLAB [41]. The performances of the developed GEP models were evaluated using the coefficient of determination ($R^2$), the root mean squared error (*RMSE*), and the mean average error (*MAE*) (21-23), by applying the following equations:

$$R^2 = \frac{\sum_{i=1}^{n} \left(h_i - \bar{h}_i\right)\left(t_i - \bar{t}_i\right)}{\sqrt{\sum_{i=1}^{n}\left(h_i - \bar{h}_i\right)^2 \cdot \sum_{i=1}^{n}\left(t_i - \bar{t}_i\right)^2}}, \tag{2}$$

$$RMSE = \sqrt{\frac{\sum_{i=1}^{n}(h_i - t_i)^2}{n}}, \tag{3}$$

$$MAE = \frac{1}{n}\sum_{i=1}^{n}|h_i - t_i|. \tag{4}$$

In these equations, $h_i$ and $t_i$ are measured and predicted output ($C_c$) values, respectively, for the *i*th data point. Furthermore, $\bar{h}_i$ and $\bar{t}_i$ are averages of the measured and predicted values, respectively, and *n* is the number of samples [28,29].

*3.3. Model Development*

The database was divided into two subsets in order to avoid an overfitting issue, a training subset and a validation subset. The GEP-based model was trained using the training subset, while the validation subset was used for validating purposes and for avoiding overfitting [34]. The final model (prediction equation) was selected based on model simplicity and the performances of the training and validation subsets. Performance criteria were based on the highest $R^2$ and lowest *RMSE* and *MAE* of the training and validation subsets. After training, the candidate models were applied to the unseen validation subset to ensure their good performance. The proportion of training to validation subset sizes with respect to the whole data is commonly selected as 60%–75% and 25%–40%, respectively. In the current study, 75% (81 data points) and 25% (27 data points) of total data points were assigned to the training subset and validation subset, respectively.

The GEP algorithm was executed several times with a varied combination of influential parameters in order to identify the best model. This process was based on values suggested by previous works [31,32,34]. Table 2 includes the parameters of various runs. Reasonably large numbers were considered for size of population and generations to guarantee that optimal models were achieved. In the developed GEP-based model, individuals were identified and transferred into further generations based on a fitness evaluation carried out with roulette wheel sampling, considering elitism. Such an evaluation could guarantee successful cloning of the best individual. Furthermore, variations in the population were carried out through genetic operators on the chosen chromosomes, including crossover, mutation, and rotation [10].

**Table 2.** Parameters used for implementation of the GEP-based model.

| Parameter | Setting |
|---|---|
| Number of chromosomes | 50 to 1000 |
| Number of genes | 3 |
| Head size | 8 |
| Tail size | 17 |
| Dc size | 17 |
| Gene size | 42 |
| Gene recombination rate | 0.277 |
| Gene transportation rate | 0.277 |
| Function set | +, −, ×, /, exp, ln, and Inv |

In every GEP-based model, the values of the setting parameters have a significant impact on model performance. These parameters include the quantity of genes and chromosomes, in addition to a gene's head size and the rate of genetic operators. Since minor information was available about GEP parameters in the literature, appropriate settings were selected based on a trial and error scheme (see Table 2).

Furthermore, to facilitate the development of the GEP-based model, the following closed-form equation was developed and utilized:

$$C_c = e_0 + \left[\frac{e_0 + 2LL}{e_0 - 6.87}\right] \times \left[-0.35 + LL^2\right] + \left[\log(2e_0 + 2LL - 2PL + 0.15)\right]^2. \tag{5}$$

Figures 6–8 present the measured values of $C_c$ obtained from laboratory experiments versus predicted values. These figures represent the measured values versus predicted values for the training subset, validation subset, and entire set, respectively. Furthermore, Table 3 summarizes the GEP-based model performance in terms of $R^2$, *RMSE*, and *MAE* for these sets. Smith [42] has stated that for a coefficient of determination of $|R| > 0.8$, a strong correlation exists between measured and predicted values. Based on Table 3, the developed GEP-based model had a high $R^2$ for the training subset,

validation subset, and entire dataset. In addition, the model exhibited a relatively low *RMSE* and *MAE* for all of these sets.

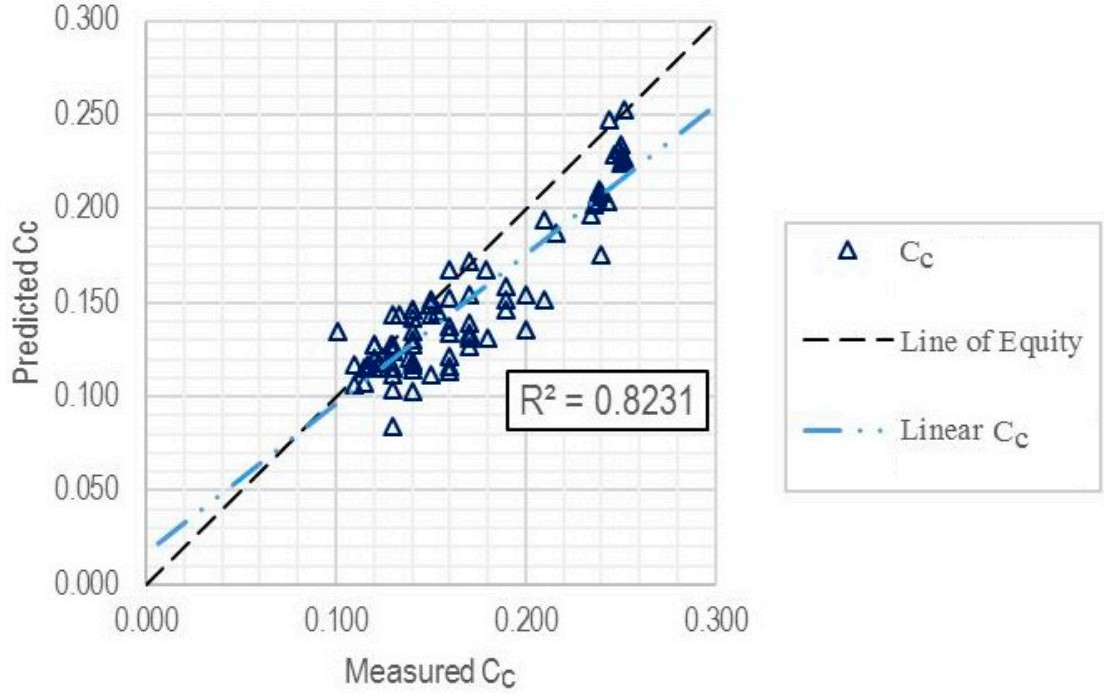

**Figure 6.** Predicted versus measured $C_c$ for the training subset.

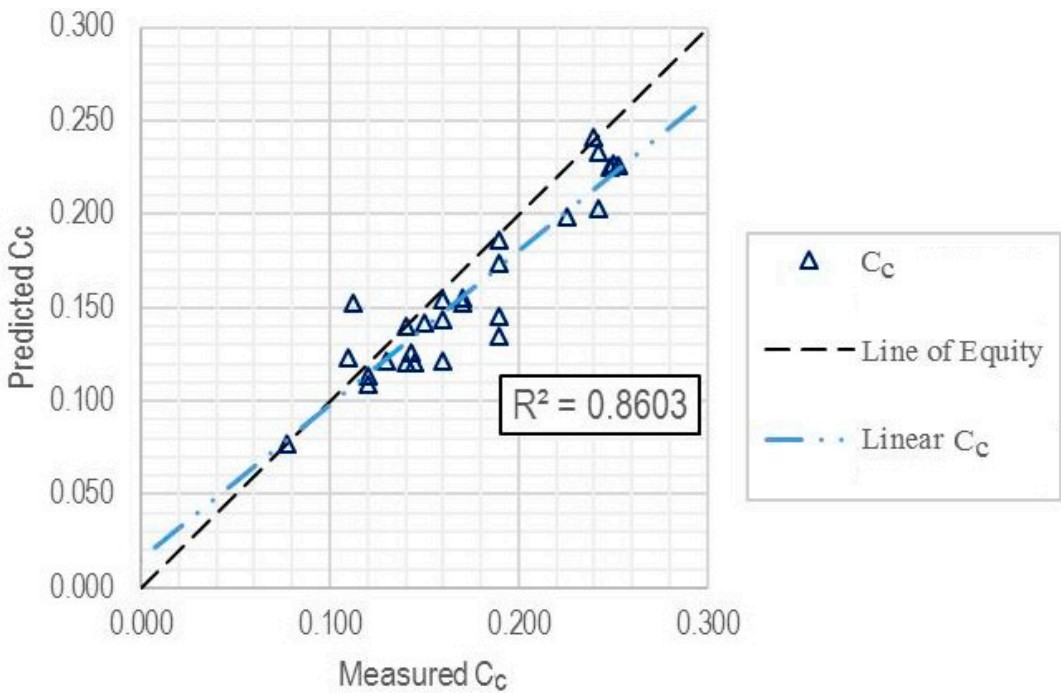

**Figure 7.** Predicted versus measured *Cc* for the validation subset.

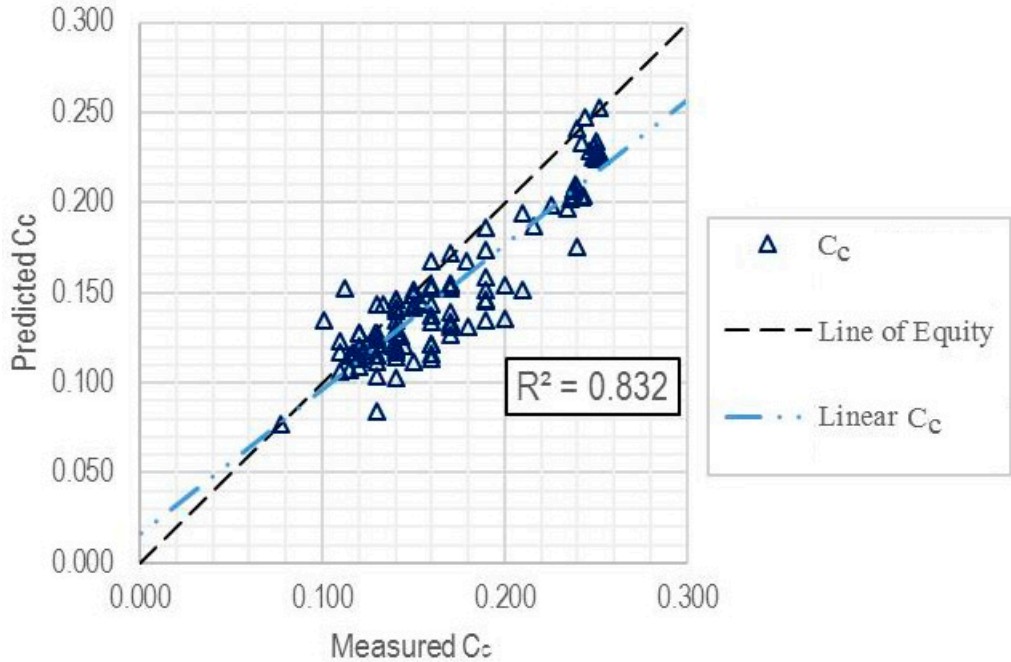

**Figure 8.** Predicted versus measured $C_c$ for the entire dataset (training + validation).

**Table 3.** Model performance. RMSE: root mean squared error; MAE: mean average error.

| Set | Number of Data Points | $R^2$ | *RMSE* | *MAE* |
|---|---|---|---|---|
| Training subset | 81 | 0.8231 | 0.0269 | 0.0213 |
| Validation subset | 27 | 0.8603 | 0.0237 | 0.0189 |
| Entire dataset | 108 | 0.8320 | 0.0262 | 0.0207 |

*3.4. Additional Evaluation of Model Performance*

In this section, the performance of the developed GEP-based model is evaluated based on various statistical parameters found in the literature. These statistical parameters, along with their acceptance criteria, are presented in Table 4. The parameters used in this table are all as previously defined. Furthermore, the developed model was evaluated based on these statistical parameters, and the results are presented in this table. As can be seen in Table 4, the developed model met all of the criteria for additional statistical parameters, revealing the decent performance of the proposed model.

**Table 4.** Evaluating the developed GEP-based model using additional statistical parameters.

| Statistical Parameter | Source | Criteria | Evaluation for GEP-Based Model |
|---|---|---|---|
| $k = \frac{\sum_{i=1}^{n}(h_i \times t_i)}{h_i^2}$ | Golbraikh and Tropsha [43] | $0.85 < k < 1.15$ | 1.001 |
| $k' = \frac{\sum_{i=1}^{n}(h_i \times t_i)}{t_i^2}$ | Roy and Roy [44] | $0.85 < k' < 1.15$ | 0.989 |
| $R_m = R^2 \times \left(1 - \sqrt{R^2 - Ro^2}\right)$ | Roy and Roy [44] | $0.5 < R_m$ | 0.503 |
| $Ro^2 = 1 - \frac{\sum_{i=1}^{n}\left(t_i - h_i^o\right)^2}{\sum_{i=1}^{n}\left(t_i - \bar{t}_i\right)^2}, h_i^o = k \times t_i$ | Roy and Roy [44] | Should be close to 1.0 | 1.000 |
| $Ro'^2 = 1 - \frac{\sum_{i=1}^{n}\left(t_i - t_i^o\right)^2}{\sum_{i=1}^{n}\left(h_i - \bar{h}_i\right)^2}$ | Roy and Roy [44] | Should be close to 1.0 | 0.998 |

Table 5 presents a comparison of the developed GEP-based model to previous models found in the literature. The previous models consist of either regression-based equations or robust AI methods,

such as MEP, ANNs, or MGGP. It is worth mentioning that these AI methods do not provide any closed-form solution. The AI methods had a relatively high $R^2$, mainly due to their black-box nature of connecting inputs and outputs. Nevertheless, the developed GEP-based model had a higher $R^2$ compared to the existing AI methods. However, MEP, ANNs, and MGGP had a lower error in terms of *RMSE* and *MAE*.

**Table 5.** Performance comparison of the current developed GEP-based model to existing models. MEP: multi-expression programming; ANN: artificial neural network.

| Source | Model Description | Performance Measure | | |
|---|---|---|---|---|
| | | $R^2$ | *RMSE* | *MAE* |
| Skempton [8] | Regression equation | 0.367 | 0.072 | 0.056 |
| Nishida [6] | Regression equation | 0.752 | 0.301 | 0.285 |
| Cozzolino [4] | Regression equation | 0.752 | 0.105 | 0.103 |
| Terzaghi and Peck [9] | Regression equation | 0.367 | 0.110 | 0.077 |
| Azzouz et al. [3] | Regression equation | 0.752 | 0.036 | 0.032 |
| Mayhe [5] | Regression equation | 0.367 | 0.102 | 0.073 |
| Park and Lee [7] | ANN | 0.752 | 0.089 | 0.085 |
| Mohammadzade et al. [28] | MEP | 0.811 | 0.019 | 0.016 |
| Mohammadzade et al. [29] | ANN | 0.859 | 0.017 | 0.014 |
| **Current Study: the proposed model** | **GEP** | **0.832** | **0.026** | **0.021** |

Based on Table 5, the developed GEP-based model outperformed the regression models, since the regression models considered only a small quantity of base functions. Therefore, such models could not be used for the complex interactions of soil parameters (i.e., *LL*, *PL*, and $e_0$) and $C_c$. However, the developed GEP-based model considered a variety of base functions and their combination in order to achieve a closed-form equation with high performance. The developed GEP-based model directly considered the experimental data with no prior assumptions. In other words, contrary to traditional regression models, GEP did not assume any predefined shape for the solution equation. The high values of $R^2$ presented in Table 5 indicate that the developed GEP-based model was very successful at fitting the measured $C_c$ to the input parameters of *LL*, *PL*, and $e_0$.

## 4. Conclusions

$C_c$ is a significant parameter in determining the settlement of fine-grained soil layers subjected to loads, such as in buildings, vehicles, and infrastructure. If $C_c$ is not estimated accurately, soil settlement is not predicted accurately. Thus, determining $C_c$ is of significant importance in settlement calculations. However, measuring $C_c$ using the traditional oedometer test method is time-consuming, needs skilled technicians, and requires special laboratory equipment. Therefore, the estimation of $C_c$ using other parameters of fine-grained soils, such as *LL*, *PL*, and $e_0$, would eliminate the time and costliness associated with the oedometer test. In this study, GEP was employed to develop a model for estimating $C_c$ using *LL*, *PL*, and $e_0$. Here, 108 data points containing $C_c$, *LL*, *PL*, and $e_0$ were used to train and validate the model. The model was developed based on tuned calibration parameters using trial and error. A closed-form solution was derived from the developed GEP-based model, which is anticipated to aid geotechnical researchers in determining $C_c$ with considerable savings in associated time and costs. This closed-form equation for predicting $C_c$ was employed to develop surface charts to predict $C_c$ based on *LL* and *PL* for a certain $e_0$.

The performance of the developed GEP-based model was evaluated using the coefficient of determination ($R^2$) and two error measures, namely root mean squared error (*RMSE*) and mean average error (*MAE*). The $R^2$ values were 0.8231, 0.8603, and 0.8320 for the training subset, validation subset, and entire dataset, respectively. In addition, *RMSE* was 0.0269, 0.0237, and 0.0262 for the training subset, validation subset, and entire dataset, respectively. A high $R^2$ and low error indicated the highly acceptable performance of the GEP-based model. Additional performance measures found

in the literature were employed to further evaluate the performance of the developed GEP-based model. This evaluation revealed that the model had a decent performance based on additional performance measures.

Contrary to the classical models for estimating $C_c$, such as regression models, the developed GEP-based model revealed highly nonlinear behavior and included a complex combination of influential input parameters (i.e., *LL*, *PL*, and $e_0$). In general, $C_c$ was positively correlated with $e_0$. Furthermore, *LL* and $e_0$ had a higher influence on the estimation of $C_c$ compared to *PL*. A comparison of the developed model to previous models in the literature revealed its good performance, which guarantees the use of this GEP-based model in practical applications.

**Author Contributions:** Review, formal analysis, methodology, modeling data curation and analyzing the results, D.M.S., A.M., and S.-F.K.; soil expertise, D.M.S., J.H.M.T., S.-F.K., and E.N.; machine learning expertise, A.M. and D.M.S.; management, conceptualization, writing, and administration A.M.; data visualization, data handling, support and data assistant; E.N.; supervision, resources, software, expertise, revision, validation and verifying the results, J.H.M.T.

**Funding:** This research received no externa funding.

**Conflicts of Interest:** The authors declare no conflicts of interest.

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
