# Peer review of "Prediction of Compression Index of Fine-Grained Soils Using a Gene Expression Programming Model"

_infrastructures, doi:10.3390/infrastructures4020026_

Round 1

Reviewer 1 Report

The study presents a prediction model for Cc on fine-grained soils by means of gene-expression programming (GEP), a biologically inspired technique to obtain closed-form solutions. 

By means of a database consisted of 108 different points, a closed-form equation solution was derived to estimate Cc based on LL, PL, and e_o.

It is an article of interest to the journal readers. but I recommend a major revision prior to its publication.

1) Major revisions

- The abstract is too extensive. I suggest reducing it to about 160-170 words.

- There are too many key words. I suggest reducing them to 4 or 5.

The parameters LL, PL and e_o should be defined, indicating what their physical or practical meaning is.

2) Minor revisions

- See attached draft.

Author Response

The study presents a prediction model for Cc on fine-grained soils by means of gene-expression programming (GEP), a biologically inspired technique to obtain closed-form solutions. 

By means of a database consisted of 108 different points, a closed-form equation solution was derived to estimate Cc based on LL, PL, and e_o.

It is an article of interest to the journal readers. but I recommend a major revision prior to its publication.

Dear Professor,

Thanks a lot for your time and your comments. I have carefully revised the manuscript and highlighted our improvements for your kind consideration. I wish that the revised version can satisfy your expectation.

1) Major revisions

- The abstract is too extensive. I suggest reducing it to about 160-170 words.

>>  The length of abstract has been reduced and  its content improves.

- There are too many key words. I suggest reducing them to 4 or 5.

>> Fixed.

The parameters LL, PL and e_o should be defined, indicating what their physical or practical meaning is.

 >> liquid limit (LL), plastic limit (PL) and initial void ratio (e0) which are three main parameters that influence Cc we defined briefly. Also we refer readers to the references [3-9] for further detailed information. The revised parts highlighted in the paper.  

2) Minor revisions

- See attached draft.

>> Thanks a lot for the guidelines on the minor revisions. We revised accordingly.

Reviewer 2 Report

The manuscript is not well-written and its idea is not novel. The reviewer regrets to reject the manuscript because of the following reasons. 

The manuscript does not have any significant contribution to the field when it was compare to the following paper: 

Mohammadzadeh, D., Bazaz, J. B., & Alavi, A. H. (2014). An evolutionary computational approach for formulation of compression index of fine-grained soils. Engineering Applications of Artificial Intelligence33, 58-68.

The writing of the manuscript should be improved as it has lots of grammatical errors. For instance, "Figs 2-5 illustrates".

Authors should give proper citations for their equations. See Eq. 1 for an example. 

Author Response

Reviewer 2.

The manuscript is not well-written and its idea is not novel. The reviewer regrets to reject the manuscript because of the following reasons. 

>> Thank you for your comments. The idea to use artificial intelligence for the prediction of compression index of fine-grained soils has recently become popular. We have previously proposed two methods to address this issue and our work is progressing to reach higher accuracy and reliability. In the future work we will be aiming at using more complex machine learning methods e.g. ensembles and hybrid machine learning models to reach better results and there is still room for improvement. 

The manuscript does not have any significant contribution to the field when it was compare to the following paper: 

Mohammadzadeh, D., Bazaz, J. B., & Alavi, A. H. (2014). An evolutionary computational approach for formulation of compression index of fine-grained soils. Engineering Applications of Artificial Intelligence33, 58-68.

>> the model which had been proposed in our former work in 2014, mentioned above, is based on multi expression programming (MEP). However, in an attempt to improve the quality of the prediction, exploring the new AI methods is essential. Therefore Gene Expression Programming (GEP) has been used which reported promising results.    

The writing of the manuscript should be improved as it has lots of grammatical errors. For instance, "Figs 2-5 illustrates".

>>Thank you for commenting on the English. We have revised the work, and a native English co-author has been added to contribute in this aspect.   

Authors should give proper citations for their equations. See Eq. 1 for an example. 

>> We have cited them again more clearly.

Reviewer 3 Report

A good quality Manuscript. Although topic is not new but application of technique is interesting. Manuscript need some improvement before publication.

Some suggestion to improve are:

Introduction: Between Line 69-70..add a paragraph about application of GEP in soil engineering (previous paper). Discuss its application along with other motives related to variables of soil mechanics.

GEP: add discussion about the software via whom it was applied. At the end of manuscript (after author contribution),code can also be provided if used via some code based software.

-Heading 3 could be about consolidation test: a brief introduction about consolidation test, process, importance may with the help of equipment or process diagram.

-How Equation 5 was developed or derived??

-How we can consider Table 5 ? all these models have been applied on the same data?? a detail discussion should be there. otherwise it is useless to compare these results.

Author Response

A good quality Manuscript. Although topic is not new but application of technique is interesting. Manuscript need some improvement before publication.

>> Thanks a lot for your positive feedback and support.  

Some suggestion to improve are:

Introduction: Between Line 69-70..add a paragraph about application of GEP in soil engineering (previous paper). Discuss its application along with other motives related to variables of soil mechanics.

>> we have added this paragraph and highlighted in the paper for your kind consideration.

GEP: add discussion about the software via whom it was applied. At the end of manuscript (after author contribution),code can also be provided if used via some code based software.

>> This paper will be linked with the MDPI Data journal where code and data are available publically.  

-Heading 3 could be about consolidation test: a brief introduction about consolidation test, process, importance may with the help of equipment or process diagram.

>> information has been added.

-How Equation 5 was developed or derived??

>> from the GEP standard model description embedded into Equation 1.

-How we can consider Table 5? all these models have been applied on the same data?? a detail discussion should be there. otherwise it is useless to compare these results.

>> All models have been applied to the same data from current studies and also our former works, i.e. [28,29]

Round 2

Reviewer 2 Report

Authors have addressed my comments. 

Reviewer 3 Report

This manuscript can be accepted in current form.